# Structure of the Granitic Pegmatite Field of the Northern Coast of Portugal—Inner Pegmatite Structures and Mineralogical Fabrics

**Cristiana Faria \*,† and Carlos Leal Gomes †**

Lab2PT—Universidade do Minho, 4710-057 Braga, Portugal; caal.gomes@gmail.com
* Correspondence: cristianamfaria@gmail.com; Tel.: +351-916-087-223
† These authors contributed equally to this work.

**Abstract:** On the coastline of Northern Portugal, metamorphic formations and pegmatites were the subject of structural analysis with the main goal of understanding Variscan kinematics and related pegmatite intrusion. This study also aims to discriminate, select and characterize relevant aspects of the structure and the paragenesis of pegmatites, well exposed as a result of coastal erosion, justifying its inclusion in the geological heritage of the Northern coast of Portugal. The pegmatite bodies show distinctive internal and external structures that are attributable to different modes of emplacement and subsequent deformation. The pegmatitic implantation in the areas of Moledo and Afife occurs in an intragneissic and perigranitic environment, for the first area, and perigneissic and perigranitic environment, for the second. In Pedras Ruivas predominates the implantation into an exo-gneissic to exo-granitic domain. The Moledo veins show evidence of multiphase open/filling, revealing positions, shapes, attitudes, sizes and internal structures that change as a function of the host lithology and host structure, but mainly due to the dilation and the cycles number of local telescoping. The structural analysis of the pegmatite bodies allows the deduction of a local fulcrum of expansion that hypothetically overlaps a hidden stock of parental granite. In Afife and Pedras Ruivas, some pegmatitic lenses are specialized and mineralized in Li, Cs and Ta, with spodumene and tantalite ± cassiterite. Spodumene occurs as giant crystals, centimetric to pluri-decimetric in length, which mark very clearly the structures of in situ or in flow crystallization inside the pegmatites (primary structures) and also the secondary structures resulting from deformation. The geometric analysis of fabrics helps the individualization of well-defined stages of progressive evolution of the deformation of the pegmatites, allowing its correlation with major D2–D3 episodes of regional Variscan deformation.

**Keywords:** pegmatites; dilation; spodumene; fabric; geological heritage

## 1. Introduction

This seeks sought to explore the geometric analysis of intra-pegmatitic fractionation structures, in a typical dilational context, aiming to contribute to the kinematic interpretation of the internal structures of the pegmatite bodies and their vein fields in intra- and peri-granitic contexts.

It also tries to establish the reliability of the use of typomorphic minerals as kinematic markers, in the study of the primary internal pegmatite fabrics, either fluidal or deformational. In fact, it can contribute to accessing the temporal and chronological correlation and influence of the different phases of regional deformation over different lithological supports, complementing the approaches that consider other mega-crystalline phases. This approach is similar to that proposed in [1].

A logical implication and corollary are the attribution of a relative age of the chronological framework of implantation of these pegmatitic assemblages in the context of the structure of the

Pegmatite Belt of the Central and Iberian Variscan Province. It will also allow the establishment of a chronological sequence of Li, Cs and Ta (LCT) mineralizations, at least in this sector of the Variscan Orogene. As an associated goal and application, the study tries to discriminate, select and characterize relevant aspects which are well exposed as a result of coastal erosion, suitable for inclusion in the regional geological heritage considering the structure and the paragenesis of pegmatites.

The intertidal pegmatite and gneiss outcrops reveal some astonishing peculiarities of lithium bearing pegmatites in a situation or statute of land management that does not allow mining.

However they are extremely interesting in what concerns good representative values (mineralogical, petrological, geomorphological and archaeological) and good exposure, suitable for inclusion in the set of sites of geological interest, with didactic and touristic importance, attributed to the geological heritage of the Northern coast of Portugal [2–6].

In the coastal zone between Caminha and Viana do Castelo, a network of aplite-pegmatitic bodies do occur, being more concentrated in a gneissic host-rock in the Northern Sector, especially in the key location—Moledo. In the metasedimentary host-rocks that predominate to the south of the mentioned range, the sill and dyke network is much less dense and its conformation is more dependent on shear deformation, tangential to transcurrent.

It is known that the position, shape, orientation, and to some extent, the size of pegmatitic bodies, are controlled by a complex interaction between pegmatitic fluid pressure, rheological state of the host rock, lithostatic and directed strain, porous pressure, anisotropies of the lodging lithologies and dilatational directions.

In the upper crust, where fragile deformation conditions prevail, the combination of lithostatic and directional stresses with preexisting anisotropies of resistance to deformation, such as fractures, cleavages, pressure shadows or even stratifications, give rise to preferential attitudes of minimum strength. The pegmatites that occur in these circumstances tend to be tabular and have a preferential orientation, according to those directions.

At deeper levels of the crust, generally characterized by states of ductile deformation, pegmatitic intrusions usually assume irregular morphologies.

According to [7], the modification of the forms and attitudes of the pegmatites can be interpreted using theoretical models developed for the different combinations of the factors referenced above. Considering the differentiation of evolutionary stages of the implantation of parental plutons, the diversity of veins attitudes can interpreted by adopting the conceptual matrix of Phillips [8,9] and Roberts's [10] vein setting, which explains the distribution and attitude of tabular to lenticular bodies around circumscribed plutonic stocks.

In conjunction with these reference models, originally adjusted to the sequence and hierarchy of attitudes of intrusive tabular bodies in anorogenic (permissive and isotropic) subvolcanic environments, a lateral and apical expansion of the parental plutonisms explains, more generally, the geometries of the pegmatites that cross the contacts between granitoids and hosting formations. Pegmatites assume the configuration of sills with subparallel directions to the contacts between granite and hosting lithologies. In distal locations, they are horizontal. In proximal positions, they show attitudes of deep with a centripetal trend (for proximal sills and cone-sheet in peri-granitic swarms).

In areas near parental massifs, the preponderant effect of the configuration of the magmatic chamber in distal sectors and a greater influence of the tensile regional field, help to better understanding of the set of strikes and deeps.

In the Northern Coast Pegmatitic Field (NCPF), the positions and attitudes of the proximal pegmatite groups adjust better to the Brisbin models for medium-depth intrusion in ductile/fragile conditions, under the major influence of the local stress field [7]. The arrangement of the sets of pegmatites in a distal situation suggests a greater preponderance of the field of regional tensions [11].

The pegmatites identified were classified into two sets, according to the criteria of [12]. Some are characterized by the presence of beryl and columbite; the remainder are included in the LCT family specialized in lithium, cesium and tantalum, having spodumene or spodumene and tourmaline [11].

The former have a predominantly aluminous and potassic geochemical feature and the latter are sodium–lithium hyperaluminous. They organized into sets that can be designated as groups, according to [13].

Some show early crystallization of spodumene in large crystals, with an extension following the c-axis of a few centimeters to several decimeters and centimetric width in a and b. The habit of the crystals is prismatic elongated and, therefore, allows registering of all the variations and local displacement, as a consequence of the action of the fields of regional strength that eventually affect the pegmatites' evolution.

It is therefore reasonable to use spodumene as a mineralogical and geometric marker to estimate the progressive displacement and deformation during the intrusion of the pegmatites, from early dilation stages to deformation that overlaps the later crystallization. The fact that the genesis of spodumene is early in the paragenetic sequence potentiates its use as a marker of deformation.

## 2. Geological and Tectonic Framework

In the geological framework of the studied areas, we used the information in sheet 1 of the geological map of Portugal at the 1:200,000 scale [14] and Caminha geological map—1C at the 1:50,000 scale, reviewed in [15].

On the Northern coast of Portugal, according to [16], the structure of the Iberia Variscan Chain shows three main folding phases (D):

- 1st folding phase (D1) is marked by folds with vertical axial planes and a metamorphic surface, with the same attitude, established in a ductile regime;
- 2nd folding phase (D2), characterized by the generation of recumbent folds, associated with significant tectonic transport, laminating the previous structures, especially at the inverse fold flanks and a metamorphic surface, with the same attitude, established in a ductile regime;
- 3rd folding phase (D3), generation of folds with subvertical axial planes, with a trajectory subparallel to the Ibero-Armorican Arc, occasionally producing a metamorphic flow and axial plane surface, also very inclined to vertical.

According to Lotze's [17] tectonostratigraphy and paleogeography, modified by [18] and [19], this area is included in Central-Iberian Zone (CIZ) and Galicia-Trás-os-Montes Zone (TMGZ) (Figure 1a). The first geotectonic unit is represented by autochthonous terrenes; the second is constituted by parautochtonous terrenes; both are Paleozoic. According to the same authors the evolution of these terrenes depended on the geodynamic evolution of Iberia, with the following major episodes [16,20]:

- Opening of the Rheic ocean, between Gondwana and Avalony—Cambrian-Ordovician (500–470 Ma);
- Subduction of the SE margin of the Rheic ocean associated with the retro-arc opening of the ocean Paleothetys—Ordovician to the Silurian (430–390 Ma);
- Separation of Armórica/Iberia, by opening of Paleothetys—distensive context;
- Closing of the oceans Rheic and Paleotethys (Low to Medium Devonian, 390–370 Ma) and collision due to the Variscan Orogeny, between Iberia and Armorica;
- Collision of Avalonia with Gondwana—end of Variscan Orogeny.

The high-resolution cartography of the Northern coast is in good agreement with its inclusion in Central-Iberian Zone of the Variscan chain, considered by [21] as a Central-Iberian Pegmatite Belt in the sense of [13].

According to [21], the Central-Iberian Zone of the Variscan chain includes a great diversity of compositional and structural types of pegmatites that were implanted from endo-granitic to exo-granitic positions and are genetically related, more or less clearly, with the different types of Variscan granites.

In the NCPF, the pegmatites are distributed from endo-gneissic, such as in the Moledo area, where the hosting occurs in gneissic units, to exo-gneissic positions, such as in Pedras Ruivas (Viana do Castelo), where the pegmatites occur inside a quartzite unit surrounded by chiastoltic mica schists.

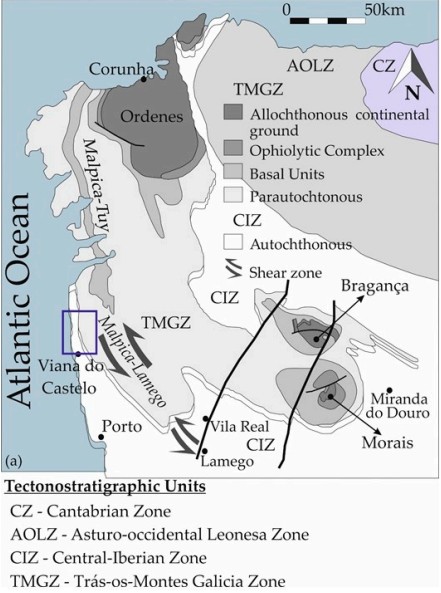 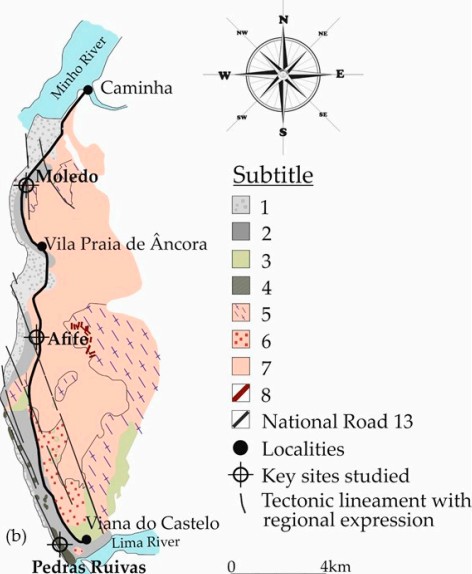

**Figure 1.** Location and orogenic setting of pegmatite key sites submitted to detailed structural analysis. (**a**) Structural arrangement of North-western Iberia (adapted from [16]); (**b**) geological sketch map modified from [14]. Legend completed in Table 1.

**Table 1.** Legend of the geological map of Portugal at the 1:200,000 scale [14] and Caminha geological map—1C at the 1:50,000 scale, reviewed in [15], applicable to Figure 1.

| Lithologies in Cartographic Sketch According to [14]. | Identified Lithologies and Adopted Designations on Sheet 1-C Reviewed by [15]. | |
| --- | --- | --- |
| | Moledo | Afife e Pedras Ruivas |
| 1. Recent and Holocene—deposits of dunes and beaches—sand or gravel. | | |
| 2. Old quaternary—river and marine deposits. | | |
| 3. Cambrian—alternation of carbonaceous phyllite and siltite. | Tourmalinites—prototurmalinites and contact tourmalinites. | Layered schists to chiastolític metapsamytes Tourmalinites—prototourmalinites and contact tourmalinites. |
| 4. Lower Ordovician—quartzites, schists and quartz matrix, conglomerates. | Quartzite to metaquartzphyllite with fissility and gray levels with ripple marks and bilobites. | Quartzite to metaquartzphyllite with fissility and gray levels with ripple marks and bilobites. |
| 5. Two mica granites sin—F3, fine grained, porphyritic. | | Porphyritic granite of Carreço |
| 6. Two mica granites sin—F3, coarse grained. | | |
| 7. Two mica granites sin—F3, medium to fine grained. | Gneisses with schlierenitic, supermicaceous, with restitic muscovite-biotitic, occasionally with garnet and tourmaline. | Gneisses with schlierenitic, supermicaceous, with restitic muscovite-biotitic, occasionally with garnet and tourmaline. |
| 8. Pegmatites | Granitic pegmatites of the class with rare metals, LCT family with spodumene and tantalite-columbite, beryl and amblygonite. | Granitic pegmatites of the class with rare metals, LCT family with spodumene and tantalite-columbite, beryl and amblygonite |

## 3. Methods

Methodologically, the development of the present study contemplates the following procedures and stages:

- Detailed geometric analysis (high resolution cartography) of pegmatites and internal, linear and planar fabrics of spodumene;
- Geometric analysis of the variability of the pegmatite inner structure and the fabric of spodumene lodged therein, in distinct sectors of NCPF, which are considered to be representatives of this variability—the Moledo to Afife Sector to the North, where the pegmatites are hosted in two mica gneisses with tourmaline and garnet, and the Pedras Ruivas Sector, in the southern part, where the pegmatites were implanted in quartzites and meta-quartzphyllite (Figure 1 and Table 1);
- Interpretation of structures resulting from imposed and superimposed deformation, from the point of view of displacement kinematics, considering the rheological behavior of host rocks in different deformational regimes.

## 4. Structural Analysis of the Pegmatites Network

### 4.1. Moledo

The Moledo site was considered as representative of the pure dilational lodging and so was considered suitable to a systematic survey of the internal pegmatitic structures characterized by the fabrics of mega feldspar, tourmaline, garnet and, rarely but more conspicuously, spodumene. This detailed geometric analysis was guided by photographic images (aerial views of vertical incidence) obtained through short-haul flights of an unmanned aerial vehicle (UAV or drone).

The integration of the two approaches allowed the correlation between internal structures and modes of intrusion. The contrasting conformation in dikes or sills was explained, comparing domains of differentiated dilation that are located in Figure 2.

The relationships between the extension (E) and the pitch (P) (measured in meters), and the E/P ratios, presented in Figure 3, show that the conditions most favorable to the diversification of internal structures occur when the magnitude of expansion is maximum, i.e., when the thickness of the in situ differentiation is larger and, especially when E/P decreases, a tendency that culminates in the fulcrum of maximum expansion that is indicated in Figure 2. In fact, structural arrangements, indicating a maximum of internal evolution, such as line-rocks or pillow-type structures and the more developed comb-structures, occur precisely in these situations. According to Figure 3, there appears to be an extensional magnitude from which the in situ and in flow crystallization differentiates itself.

In a vast set of veins whose internal geometry was analyzed, comb-type structures prevail in a number of one to three dilational/coaxial reactivations. Thus, it can be deduced that during the consolidation, the in situ crystallization processes predominated. For greater percentages of extension (>40%) and larger dilational volumes, conditions of occurrence for more than three dilational reactivations are gathered, and in addition to comb structures, line-rocks, albite-pillows and convoluted linear and planar flow fabrics do occur (Figure 4). In addition to in situ crystallization, structures with crystallization in flow were observed.

Also from the detailed geometric analysis of the Pegmatitic Group of Moledo, the possibility of defining planimetric domains with differentiated dilation arises. In fact, the pure dilation components are much more materialized than the dilation associated with shear. Consequently, the typological differentiation of the pegmatites may be related to kinematic criteria.

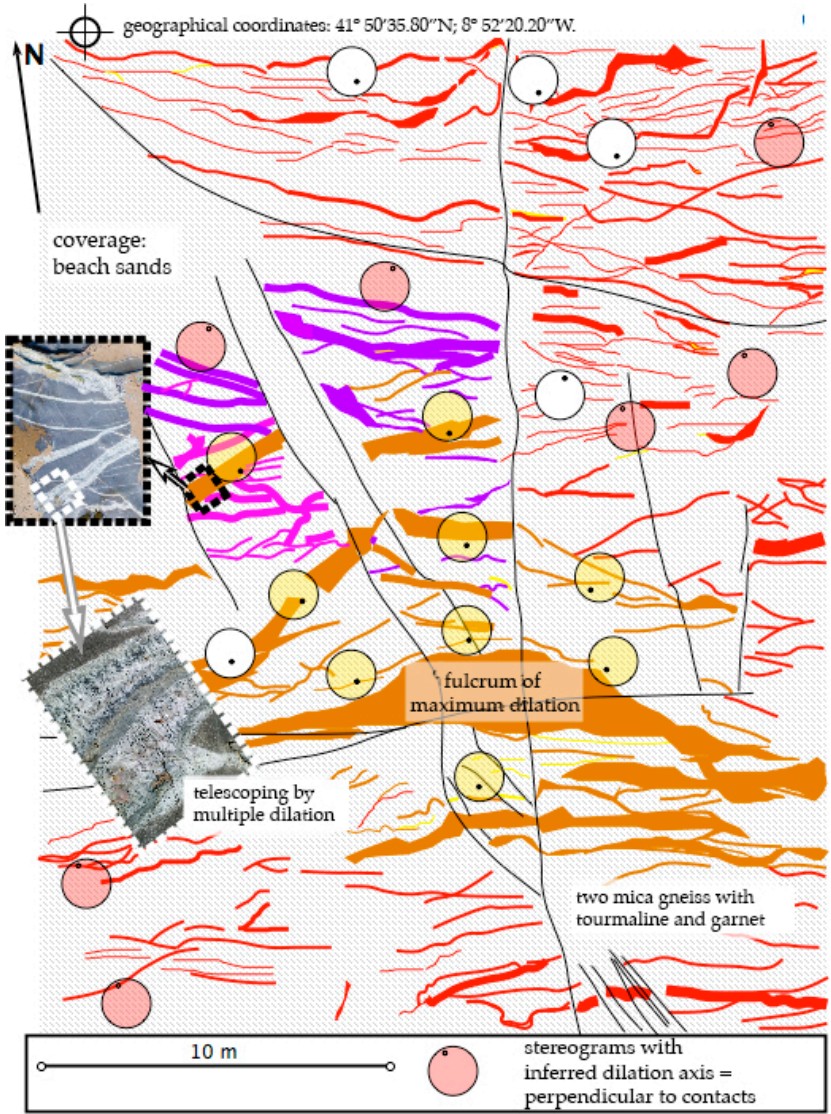

**Figure 2.** Interpretation of high-resolution cartography at the Moledo Pegmatite Group and zoning of distinct dilation domains. Different colors distinguish magnitudes and number of dilation stages.

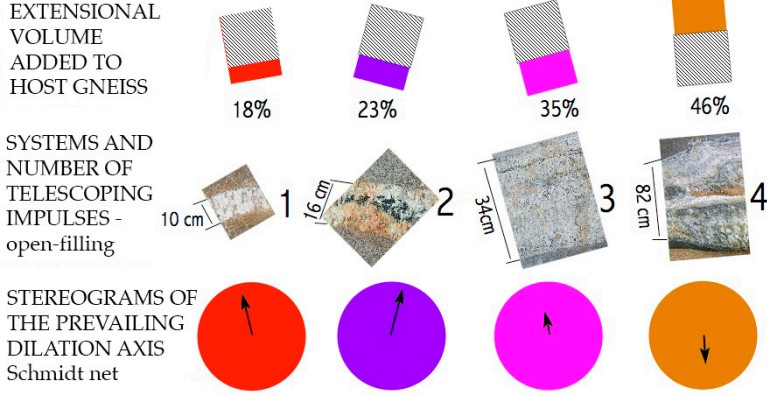

**Figure 3.** Dilation analysis—magnitude and multistage character. Colors identify the outcrops represented in the high-resolution map in Figure 2.

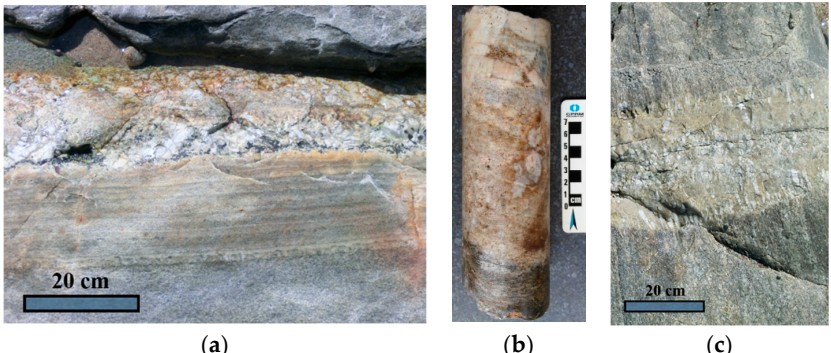

(**a**)                  (**b**)                  (**c**)

**Figure 4.** Structural arrangements and fabrics of internal fractionation in situ and in flow (in this case, essentially gravitational) resulting from different numbers of coaxial dilation pulses. (**b**) drill core of a sill polar structure (the drill core was obtained at the border of the pegmatite field). (**a**) gravitational polar structure in sill-comb structure to the roof and line-rock to the foot-wall; and (**c**) Symmetrical structure in dikes-comb layering.

Such a system reflects the diversification of internal structures in function of magnitude and time of rock-volume expansion and the prevalence of progressive or multistage dilation (Figure 5) relative to shear dilation at different regimens and alternating repeated dilation/shortening textures (Figure 6).

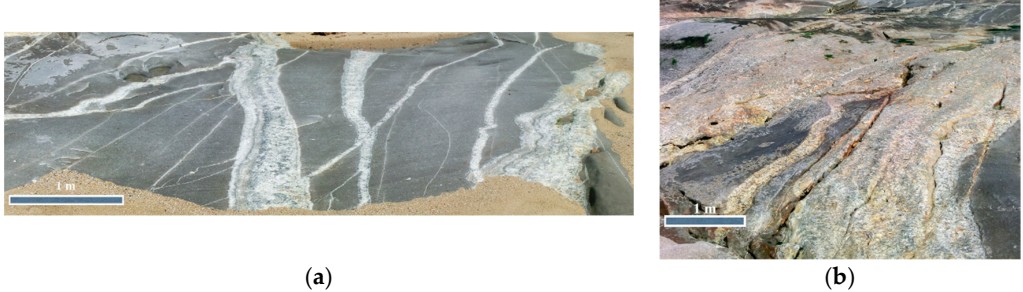

(**a**)                                (**b**)

**Figure 5.** Geometric and mineralogical expressions of different stages and magnitudes of dilation—progressive or multistage dilation structures in contrasting dilation domains. (**a**) Defined or progressive monotypic dilation, two stages—independent vein fills; and (**b**) Polytypic dilation multiple stages—common or coalescing vein fills.

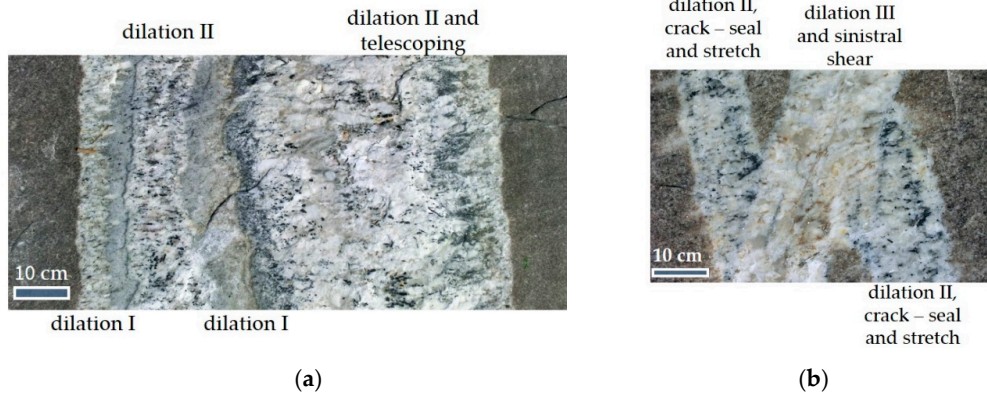

(**a**)                                (**b**)

**Figure 6.** Typology of dilatational pegmatite stages—diversity of internal structures associated with different reactivations: (**a**) dilation/shortening structures, dilation I and II with telescoping open/filling—mineralogical markers: orthoclase plumose and schorl stretched and segmented; (**b**) shear controlled expansion structures: dilation II—crack-seal and stretching; dilation III—dilation and sinistral shear.

### 4.2. Afife

The pegmatite set of Afife, like that of Moledo, is also hosted in gneissic rocks composed of a facies of two micas with garnet and tourmaline, to which there are sometimes associated xenoliths, leucocratic segregations and schlieren-like restites. In the Afife location, the vein emplacement is controlled by a pure dilation and the pegmatites are lodged according to permissive conditions, or instead, the dilation lodging is related with shear, which happens in pull-a-part amplifications and volumes.

### 4.3. Pedras Ruivas

The pegmatite group of Pedras Ruivas, unlike those already considered, is implanted in quartzite and quartz-phyllite host-rocks, the rheological contrast between the quartzite and quartz-phyllite unit, being determinant for its selective emplacement, with a competent response to ductile-fragile deformation in contrast with the surrounding horizons of high-fissile phyllits and shale formations, including the alluded chiastolite mica schists that appear more to the West. The attitude of the pegmatites is conditioned by the generation of volumes of expansion associated with the shear processes (Figure 7).

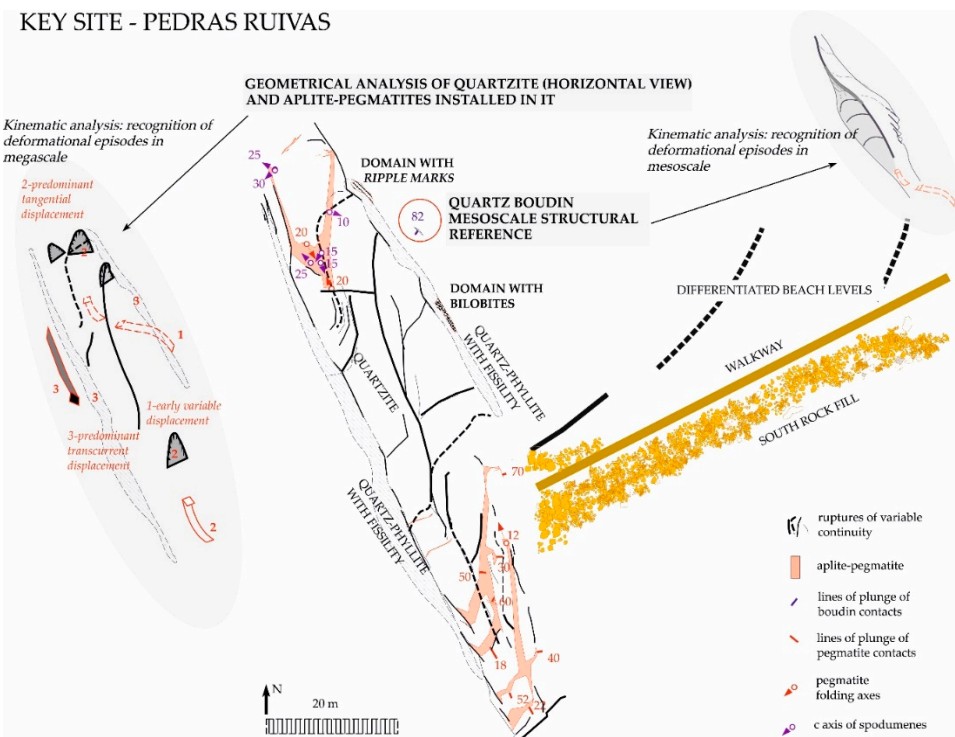

**Figure 7.** High-resolution cartography of the Pedras Ruivas Pegmatite Group and kinematic interpretation of mega and mesoscale deformational episodes.

## 5. Modes of Occurrence of Spodumene

The occurrence of spodumene in pegmatites includes the following textural architectures (Figures 8–11):

- Comb structure due to centripetal growth nucleated at the pegmatite contact;
- Dissemination of phenocrysts in a magmatic flow, highlighting the planar and linear imbrications according to the fluidalities' motion in the fine granular matrix, sometimes typically aplitic;
- Occurrence as mega to giga-crystals of sheared spodumene, due to orthogonal shortening or extension associated with shear in low viscosity state or in a ductile regime after the crystallization

of the surrounding mesostasis—spodumenes present contours according to the surrounding c/s surfaces.

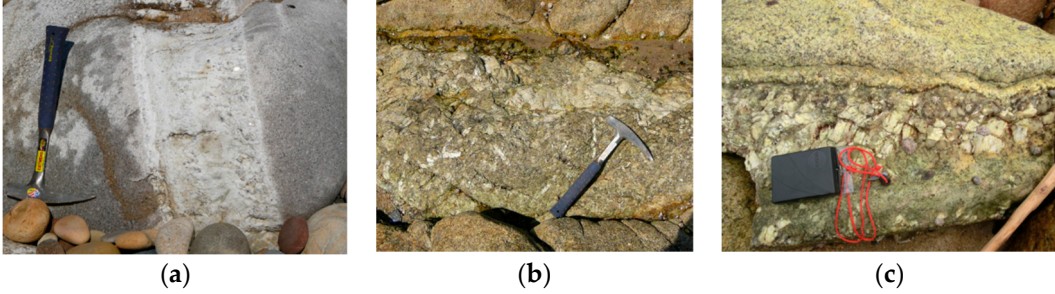

(**a**) (**b**) (**c**)

**Figure 8.** Types of pegmatites with spodumene and differentiation of tabular bodies with early comb spodumene: (**a**) dyke—narrow tabular bodies with early comb spodumene—with symmetrical inner structure; (**b**) sill—narrow tabular bodies with early comb spodumene—asymmetric polar inner structure resulting from differentiated crystallization from roof-wall to foot-wall; and (**c**) sill with same characteristics of (**b**).

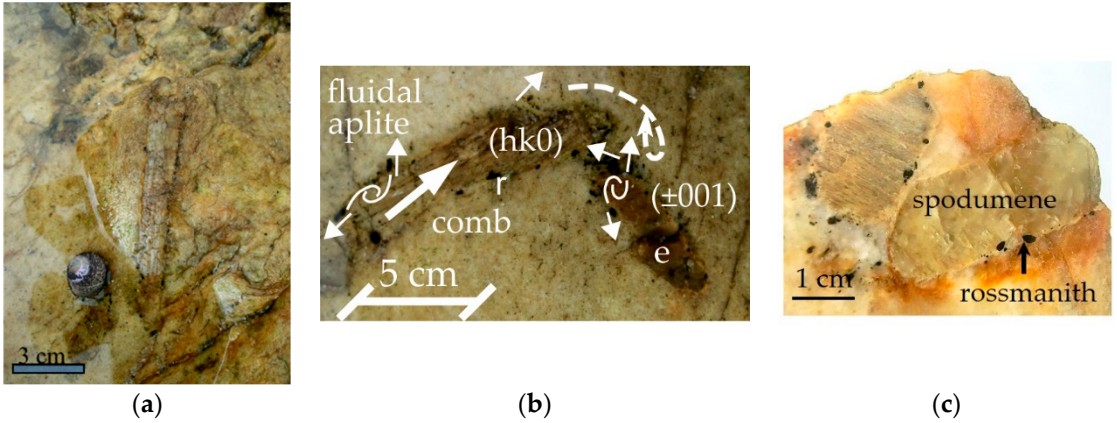

(**a**) (**b**) (**c**)

**Figure 9.** Spodumene mesoscopic fabrics—flow crystallization structure. (**a**) exposure parallel to c of the spodumene + rossmanite; (**b**) ◩ rossmanite c-axis—defines a rotational deltoid in aplitic flow—rotation under low viscosity and magmatic flow; and (**c**) cut perpendicular to c of the spodumene + rossmanite.

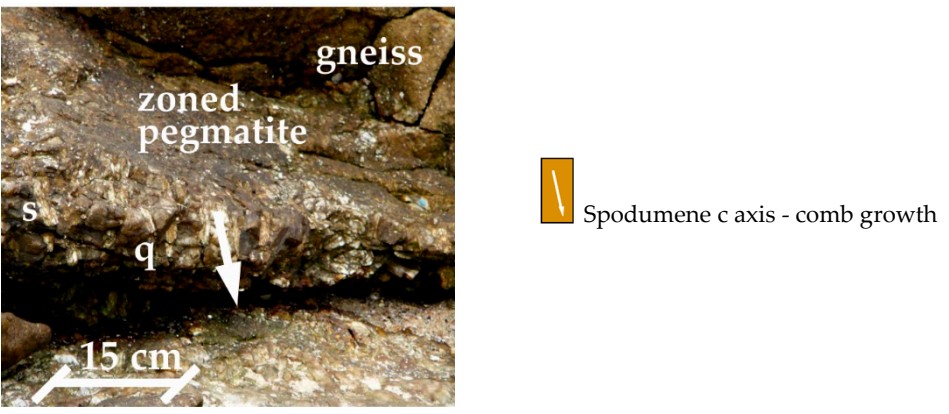

**Figure 10.** Spodumene mesoscopic fabrics–in situ crystallization structure. Pegmatite sill with comb spodumene growing from the intermediate zone to the quartz core (Afife).

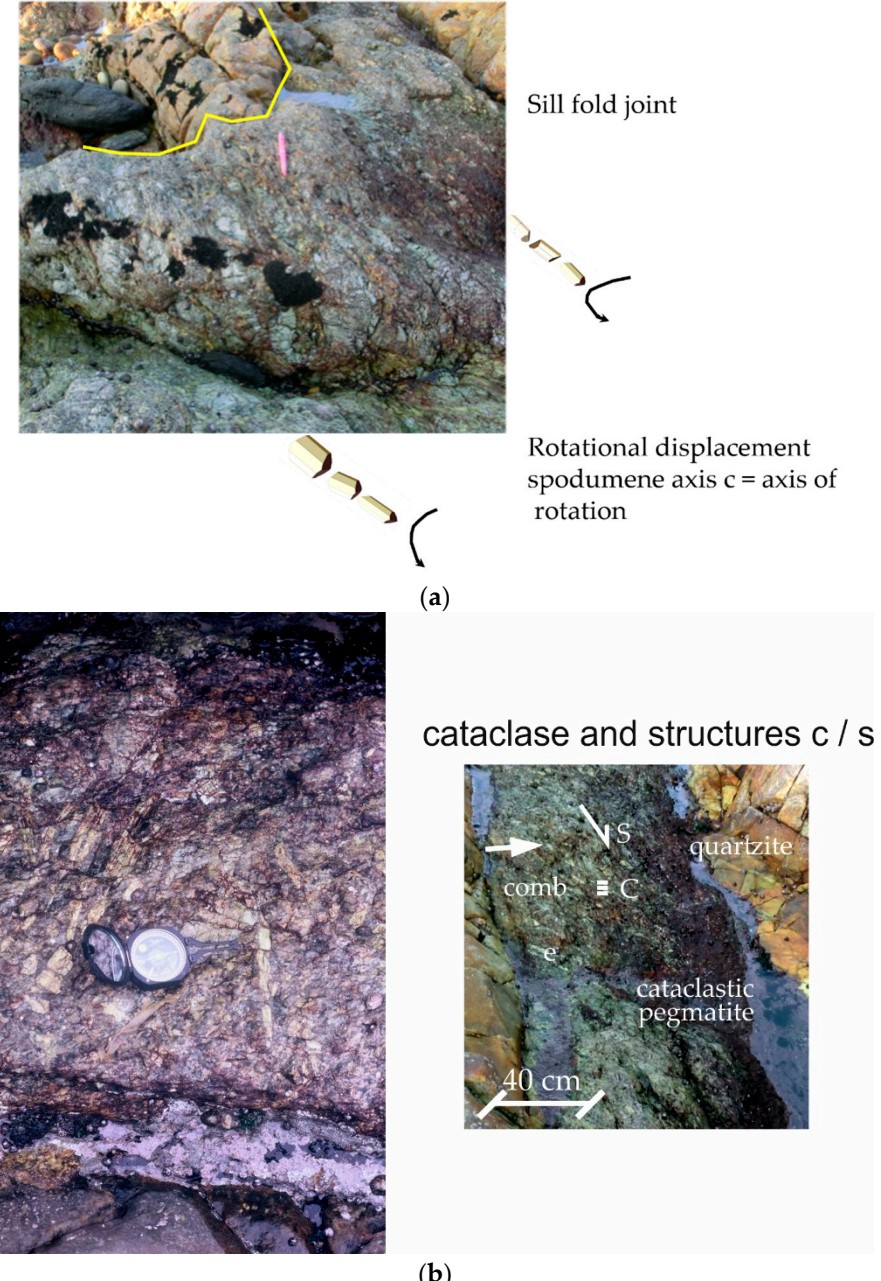

**Figure 11.** Spodumene mesoscopic fabrics (Pedras Ruivas). (**a**) Ductile imposed deformation. (**b**) Pegmatite sill hosted in low angle reverse fault with c/s and several spodumene fish-like structures.

There is geometric evidence that the early crystallization of spodumene may occur in situ or in flow (Figures 9 and 10). Deformation can be imposed to the consolidation of the host pegmatites, affecting them when they are in different states or rates of crystallization, or can be superimposed affecting them after the total crystallization or even after the metasomatism. It was possible to distinguish in structural terms the pegmatites with spodumene, attributing to this structural discrimination the following major terms of paragenetic diversity (Figures 9–11):

- Type I—narrow tabular bodies with spodumene in early comb-structures (Figure 8);
- Type II—tabular bodies with metrical width with spodumene, tourmaline, garnet and apatite-megacrysts of spodumene were observed in comb and flow structures (Figure 9);

- Type III—tabular to lenticular bodies of variable thickness, with very deformed spodumene (cataclastic to milonitic and pseudomorphic in phyllosilicates) associated with well-defined c/s (shear) structures (Figure 11);
- Type IV—lenticular bodies with spodumene units in late dilations, constituting pull-apart structures.

## 6. Discussion

The integration of the results of the detailed structural analysis allowed the differentiation of meso-scalar fabric domains, which depend very much on the different rheological behavior of the pegmatite host-rocks. The lodging of pegmatites inside gneissic rocks, in the Moledo case (Figures 12 and 13) and Afife example (Figures 14 and 15), produces inner structures clearly distinguishable from those occurring in pegmatites lodged inside quartzite strata, which is the case of Pedras Ruivas (Figures 16 and 17).

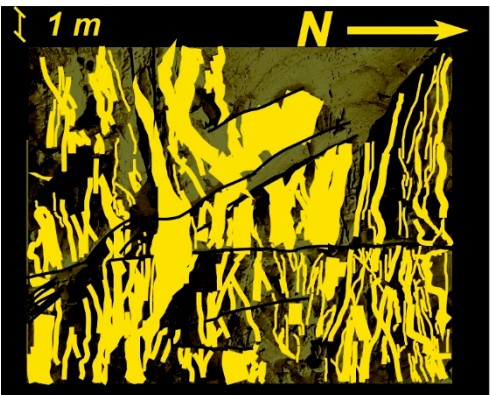

**Figure 12.** Moledo location: geometry of sills and dykes, enhanced in a vertical drone image. Morphometry and morphoscopy of the dykes (yellow bands) in the horizontal projection at sea level.

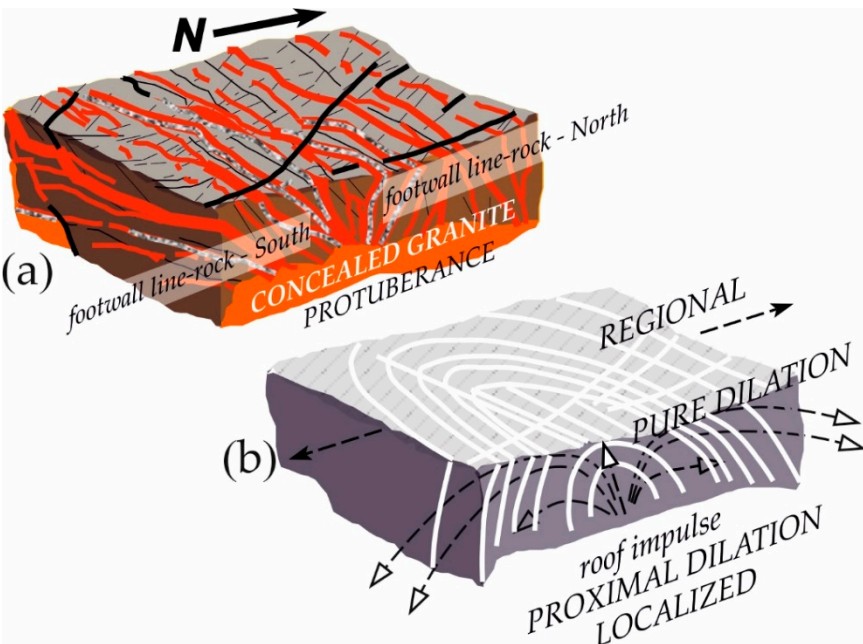

**Figure 13.** Conceptual sketch blocks for Moledo location. (**a**) Conjugated apical impulse of a parental stock, superimposed to the asymmetrical regional dilation. (**b**) Isopacs of linear dilation and anomalous magnitude responding to locally asymmetric dilation, white lines; simplified distention trajectories are represented in black dashed lines.

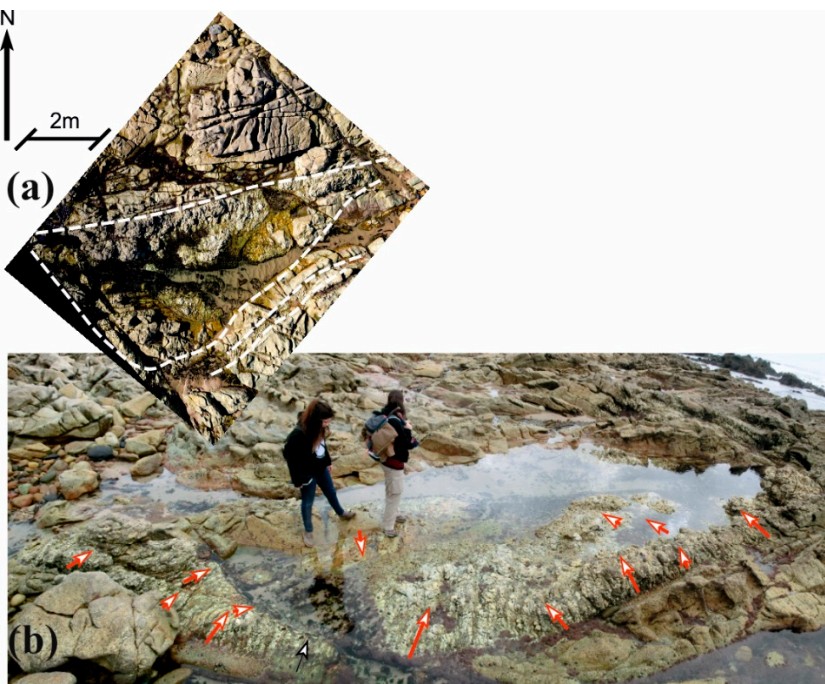

**Figure 14.** Afife pull-apart: (**a**) drone image, with delimitation of the dilation volumes; (**b**) perspective from the south, with horizontal projection of the centripetal growth vectors in the comb-structure of geminated Baveno giga-feldspars.

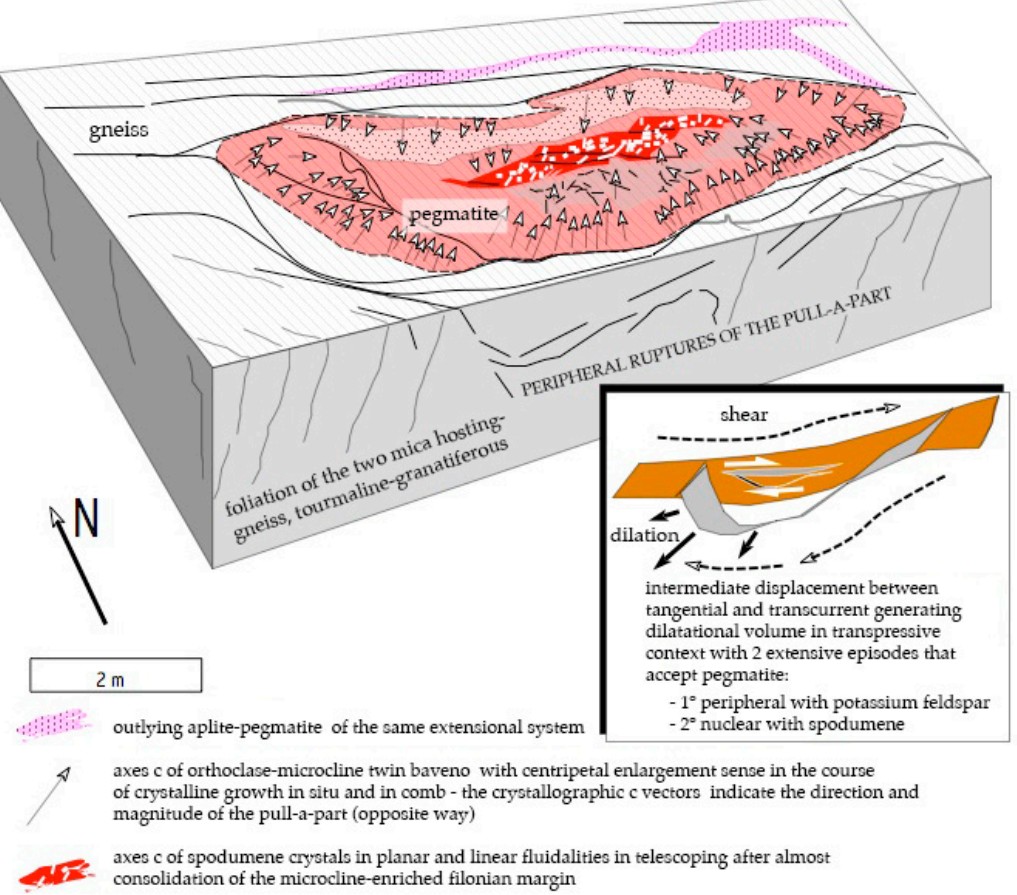

**Figure 15.** Three-dimensional sketch blocks including spodumene structures from pegmatites hosted in gneisses. The corresponding outcrop is located in Afife.

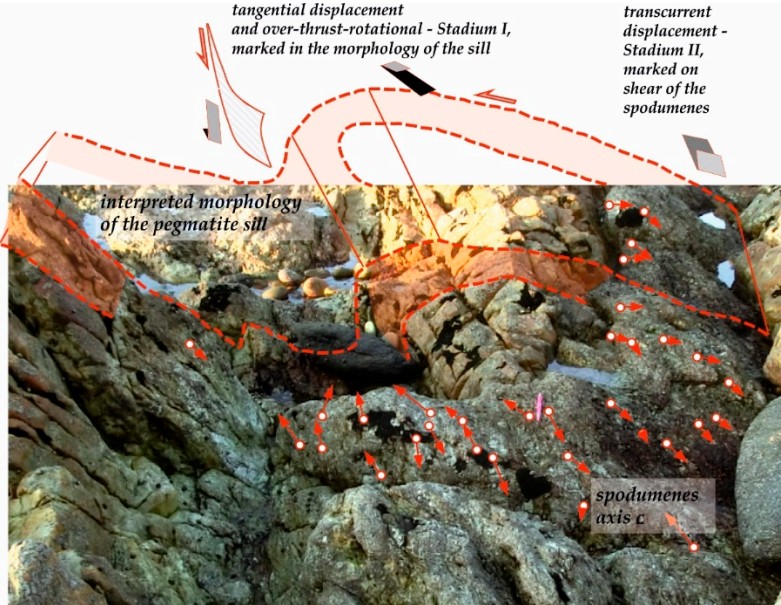

**Figure 16.** Evidence of shortening followed by transcurrent shear deformation, superimposed over the internal consolidation in a sill pegmatite body from Pedras Ruivas.

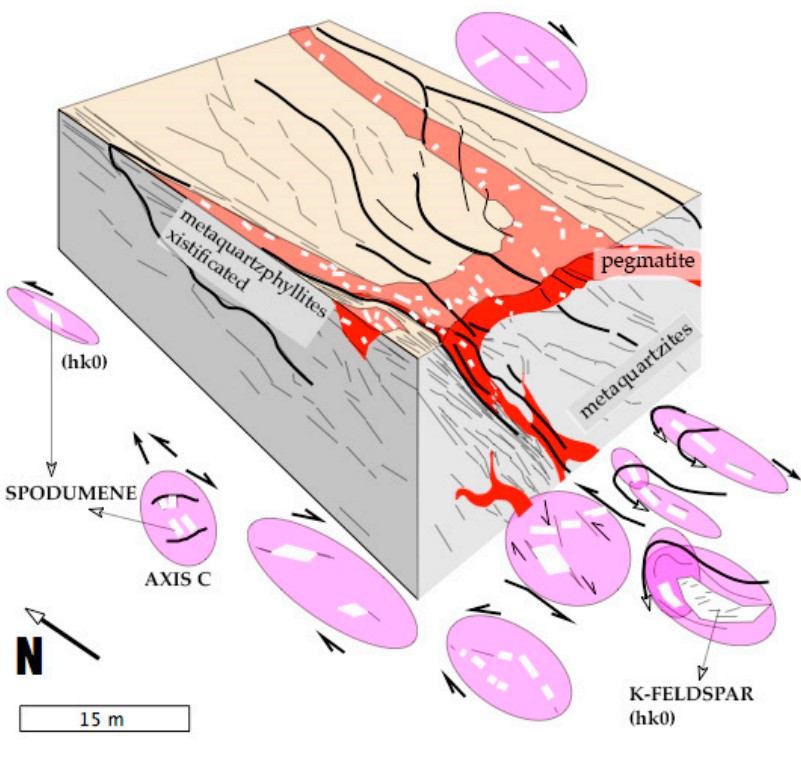

**Figure 17.** Three-dimensional sketch blocks for spodumene structures observed in pegmatites intruded in quartzite units from Pedras Ruivas. The corresponding outcrops are located in Figure 1. Note: Discontinuities attributable to the first deformation phase (D1) tend to be coincident to those related with the third deformation phase (D3).

The variation in space and time of the regional stress fields combined with the physical properties of the host-rocks had a strong influence on the structural organization of the pegmatitic bodies. Reciprocally, it allows the deduction of the main trajectories of strength, acting in the time lapse of the pegmatites' permissive emplacement (Figures 12, 13, 16 and 17).

## 7. Conclusions

The permissive emplacement of the aplite-pegmatites at the Moledo beach, marked by internal structures of in situ crystallization and centripetal fractionation, essentially depends on the dilation along an N–S axis. The most penetrative foliation of the gneisses lies between the azimuths N–S and N25° W. The dilation may have as fulcrum a circumscribed and concealed protuberance of parental granite.

In the case of the Moledo Group and possibly in many pegmatites generated in dilational environments, the diversity of internal structures varies according to the magnitude and the number of dilation episodes. The hierarchy and the attribution of a chronology for the deformation events affecting each Pegmatitic Group is influenced by the antecedent factors such as the structure of the host-rock.

The rheological behavior of the host-rock at the intrusion site has also a strong influence on the current orientation of the vein structures and on their response to overlapping deformation; consequently, the spodumene fabric and the crystal forms and shapes are affected and might register differentiated stages or variations in magnitude of the progressive deformation. In the studied region, the crystals of spodumene and the pegmatites that carry them denote the action of at least two important and independent episodes of coaxial deformation, from crystallization to evolution under subsolidus conditions.

In all pegmatites North of Pedras Ruivas, the planar-linear setting of spodumene is affected by the same deformation that generates de most penetrative foliation of the host gneisses.

At the mega scale, the results of the petrofabric study reflect the difference in rheological behavior between meta-quartzofilites and gneisses. It also indicates that the lithium mineralization in the form of spodumene is synchronous or precedes the last ductile-fragile structuring of quartzite-metaquartzphyllite of Pedras Ruivas and succeeds to the genesis of the Afife gneisses. This assumption suggests that Li mineralization can be earlier than what is generally assumed [15] and still influenced by the deformation attributed to D2 (Figure 18 and Table 2), Variscan folding phase.

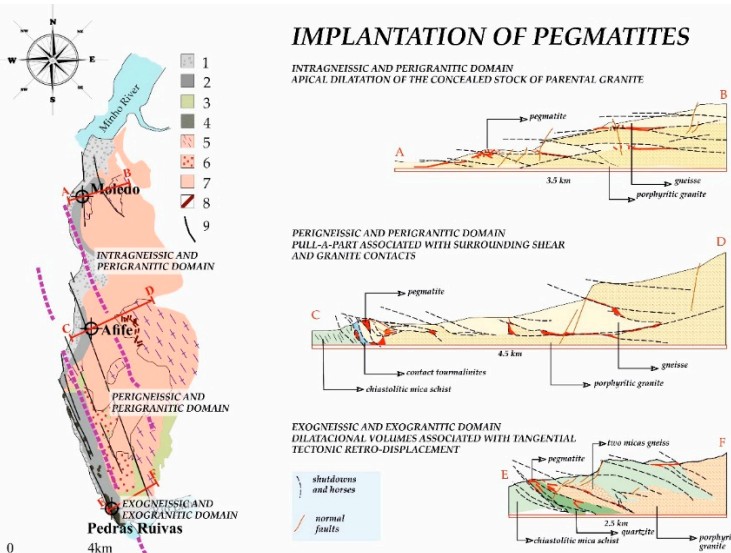

**Figure 18.** Conceptual profiles discriminating contrasting styles of pegmatite emplacement. Legend completed Table 2.

**Table 2.** Lithological correspondences between sketches of geological map and profile. Note: the tourmaline rocks and the metasomatic tourmalinites as well as the pegmatite and aplite-pegmatite outcrops do not have a definite cartographic expression at this scale. Recent and quaternary units were not included in the outline of the altimetric profiles for ease of reading.

| Lithologies in Cartographic Sketch According to [14]. | | Correspondence in the Extended Interpretative Profiles. | | |
|---|---|---|---|---|
| | | Profile AB | Profile CD | Profile EF |
| | 1. Recent and Holocene—deposits of dunes and beaches—sand or gravel. | | | |
| | 2. Old quaternary—river and marine deposits. | | | |
| | 3. Cambrian—alternation of carbonaceous phyllite and siltite. | | Chiastolite Mica schist Contact Tourmalinites | Chiastolite Mica schist |
| | 4. Lower Ordovician—quartzites, schists and quartz matrix, conglomerates. | | | Quartzite |
| | 5. Two mica granites sin—F3, fine grained, porphyritic. | Porphyritic two micas granite | Porphyritic granite of Carreço | Porphyritic granite of Carreço |
| | 6. Two mica granites sin—F3, coarse grained. | | | Granites with two micas |
| | 7. Two mica granites sin—F3, medium to fine grained | Gneiss | Gneiss | |
| | 8. Pegmatites | Pegmatite | Pegmatite | Pegmatite |

The metallogenic implications of this study led to the conclusion that the NCPF has a great importance in what concerns the geological heritage of Northern Portugal, not only due to the peculiarity and rarity of the mineralogical and pegmatitical exposures and its geomorphology and induced coastal landscape, but also because it allowed the deduction of a conceptual model of emplacement, which represents some of the oldest and well-expressed evidence of Li–Ta metallogenesis in the context of Iberia, particularly in the Central Iberian Pegmatite Belt.

**Author Contributions:** Formal analysis, C.F. and C.L.G.

**Funding:** This research received no external funding.

**Conflicts of Interest:** The authors declare no conflict of interest.

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
