# Peer review of "Structure of the Granitic Pegmatite Field of the Northern Coast of Portugal—Inner Pegmatite Structures and Mineralogical Fabrics"

_heritage, doi:10.3390/heritage2010021_

Round 1

Reviewer 1 Report

For a reader from a slightly different field, it would be helpful to once define the most specialized expressions when they are first mentioned, particularly as many expressions are not really available in google, I assume only in a geological dictionary. 

(e.g. 294 assomo?)

Author Response

Taking into account the suggested revisions we restructured the article and English was verified.

Reviewer 2 Report

The ms focuses on Structural Geology and Mineralogy issues related to a Granitic Pegmatitic field located in Northern Portugal. The ms is well structured, adequately illustrated and well organized (although the English should be checked by a native speaker as it contains several errors). However, my main concerns about this article is that it is a "Earth Science" not "Heritage" local  case study that can only be of interest to readers specifically dealing with Portuguese regional geology. As such, in my opinion, it should be submitted to a Journal related to this sector and not to Heritage.

Author Response

(The authors gave the same response as above.)
